# GWAS and Transcriptome Analysis Reveal Key Genes Affecting Root Growth under Low Nitrogen Supply in Maize

**DOI:** 10.3390/genes13091632

**Published:** 2022-09-11

**Authors:** Yunyun Wang, Tianze Zhu, Jiyuan Yang, Houmiao Wang, Weidong Ji, Yang Xu, Zefeng Yang, Chenwu Xu, Pengcheng Li

**Affiliations:** 1Jiangsu Key Laboratory of Crop Genetics and Physiology, Key Laboratory of Plant Functional Genomics of the Ministry of Education, Jiangsu Key Laboratory of Crop Genomics and Molecular Breeding, Agricultural College, Yangzhou University, Yangzhou 225009, China; 2Jiangsu Co-Innovation Center for Modern Production Technology of Grain Crops, Yangzhou University, Yangzhou 225009, China; 3Joint International Research Laboratory of Agriculture and Agri-Product Safety, Ministry of Education of China, Yangzhou University, Yangzhou 225009, China

**Keywords:** low nitrate, maize, root, genome-wide association study, NAC transcription factor

## Abstract

Nitrogen (N) is one of the most important factors affecting crop production. Root morphology exhibits a high degree of plasticity to nitrogen deficiency. However, the mechanisms underlying the root foraging response under low-N conditions remain poorly understood. In this study, we analyzed 213 maize inbred lines using hydroponic systems and regarding their natural variations in 22 root traits and 6 shoot traits under normal (2 mM nitrate) and low-N (0 mM nitrate) conditions. Substantial phenotypic variations were detected for all traits. N deficiency increased the root length and decreased the root diameter and shoot related traits. A total of 297 significant marker-trait associations were identified by a genome-wide association study involving different N levels and the N response value. A total of 51 candidate genes with amino acid variations in coding regions or differentially expressed under low nitrogen conditions were identified. Furthermore, a candidate gene *ZmNAC36* was resequenced in all tested lines. A total of 38 single nucleotide polymorphisms and 12 insertions and deletions were significantly associated with lateral root length of primary root, primary root length between 0 and 0.5 mm in diameter, primary root surface area, and total length of primary root under a low-N condition. These findings help us to improve our understanding of the genetic mechanism of root plasticity to N deficiency, and the identified loci and candidate genes will be useful for the genetic improvement of maize tolerance cultivars to N deficiency.

## 1. Introduction

Nitrogen (N) is one of the most important mineral nutrients for plant growth and development [1]. To increase crop grain yields, large amounts of N fertilizers are applied; however, only 30–50% of the applied N is absorbed and utilized by crops [2]. The excessive application of N fertilizers leads to severe environmental problems, including soil acidification as well as water and air pollution [3,4,5]. In developing countries, low-N (LN) availability is the primary factor limiting crop production [6]. Maize is a major crop cultivated worldwide, but almost 20% of the N fertilizer applied globally is for maize production [7]. Therefore, breeding new maize varieties with an improved N use efficiency (NUE) is crucial for ensuring food security and environmental sustainability.

Nitrogen use efficiency is determined on the basis of three key intrinsic factors, namely N uptake, assimilation, and remobilization [2]. In previous studies, researchers mainly focused on the shoot biomass and grain yield to improve N transport and assimilation, and several key enzymes involved in N metabolism were identified, including nitrate reductase, nitrite reductase, and glutamine synthase [2]. There has been relatively little research on the relevance of root phenes for efficient N capture [8]. Because roots are essential for the acquisition of N, improving root traits can increase the uptake of N [9]. The complex maize root system comprises an embryonic root system that includes the primary root and some seminal roots as well as postembryonic shoot-borne roots and postembryonic lateral roots that are formed by all root types [10]. Under LN conditions, maize lines with only a few long lateral roots would have greater axial root elongation, deeper rooting, and acquire more N than lines with many but short lateral roots [11]. Increasing maize root growth and development (root dry weight, root length, and root density) via genetic manipulation can increase N uptake and yield [9]. Additionally, LN conditions can stimulate root growth to increase the uptake of N. Earlier studies revealed that the limited availability of N induces the lengthening of maize axial root and lateral roots [12,13]. Moreover, the changes in the total root length [i.e., ratio of the total root length under LN and high-N (HN) conditions] and shoot N contents are significantly positively correlated under LN conditions [14].

Plants often grow in spatially heterogeneous soil in which there are temporal fluctuations in nutrient concentrations. Nitrate is the most essential N-compound in aerobic soil for plants and a crucial signaling molecule involved in lateral root development [15]. In Arabidopsis, the expression of *ARABIDOPSIS NITRATE REGULATED 1* (*ANR1*), which encodes a MADS-box transcription factor, is influenced by local nitrate levels and stimulates lateral root proliferation in nitrate-enriched zones [16]. Clearly, the NITRATE TRANSPORTER 1.1 (NRT1.1) can activate the ANR1-dependent signaling pathway that controls nitrate-induced changes in lateral root elongation [17]. In rice, miR444a targets four MADS-box genes (*OsMADS-23*, *OsMADS-27a*, *OsMADS-27b*, and *OsMADS-57*) to trigger lateral root development under low-nitrate conditions [18]. In maize, the production of the truncated MADS-box transcription factor ZmTMM1 is induced by the local nitrate supply and modulates lateral root development [19]. Previous studies revealed that the regulatory effects of nitrate on root growth involve an interaction between nitrate and the auxin signal transduction pathway [17,20,21]. Additionally, NRT1.1 modulates lateral root growth in response to nitrate by integration of nitrate signaling and auxin biosynthesis and transport [21,22]. Naulin et al. [23] have demonstrated that nitrate stimulates primary root growth via the cytokinin signaling pathway. At high nitrate concentrations, ethylene regulates lateral root development by modulating nitrate transporters in Arabidopsis [24]. Abscisic acid (ABA) also plays a central role in mediating the inhibitory effect of nitrate on lateral root formation, which also involves *ABI4* and *ABI5* [25]. Jia et al. [26,27] confirmed that changes in root elongation under LN conditions depend on brassinosteroid biosynthesis and signaling. Despite these earlier findings, the mechanisms regulating the maize root-foraging response under low-nitrate conditions should be further characterized.

Several studies indicated that variations in root architecture-related traits are associated with the nitrate transport system. The primary root length [27] and average lateral roots length [26] vary substantially among 200 Arabidopsis accessions under both LN and HN conditions. There are also considerable differences in the physiological NUE among rice varieties under varying N conditions [28]. The diversity in N-related phenotypes may be partly explained by differences in the genes encoding nitrate transporters or other associated factors. The naturally occurring genetic variation in *NRT1.1B* can affect nitrate uptake and nitrate root-to-shoot transport. For example, the *NRT1.1B*-indica variation leads to the upregulated expression of nitrate-responsive genes [29]. In Arabidopsis, noncoding variations in *DWARF1* [14] and *BSK3* [27] respectively modify brassinosteroid biosynthesis and signaling to regulate root foraging under LN conditions. A genome-wide association study (GWAS) is a powerful approach for identifying the single nucleotide polymorphism (SNP) markers associated with specific phenotypes. Natural variations in root traits under LN conditions have been detected in Arabidopsis by a GWAS [14,27]. The allelic variation that causes root trait differences under LN conditions has not been thoroughly investigated in maize. In this study, we examined the natural variations in the root traits of 213 maize inbred lines under LN conditions by performing a GWAS. The objectives of the study were as follows: (i) to identify the genetic variations that modulate root growth under LN conditions and (ii) to detect the SNPs and genes related to root traits under LN conditions.

## 2. Materials and Methods

### 2.1. Plant Materials and Growth Conditions

A total of 213 maize inbred lines collected from 5 heterotic groups (Reid, Lancaster, Tang SiPing Tou, Zi330, and mixed group) were used in this study [30,31]. The inbred lines were grown in paper roll system, as previously described [32,33]. All seeds were surface sterilized in 10% H_2_O_2_ (*v*/*v*) for 20 min and rinsed with sterile distilled water, then followed by saturated CaSO_4_ for 6 h. Seeds were placed between moistened filter paper and germinated in the dark at 28 °C for two days. Every eight uniform seedlings per genotype were vertically placed approximately 2 cm below the top edge of a two-layer wet germination roll paper (Anchor Paper Company, St Paul, MN, USA), covered with another paper and then they were rolled up. The rolls were placed upright in 39.5 × 29.5 × 22.5 cm black incubators containing 7.5 L nutrient solution, which was renewed every 3 days. The composition and concentration of the nutrient solution are as follows: 1 mM Ca(NO_3_)_2_, 0.75 mM K_2_SO_4_, 0.65 mM MgSO_4_, 0.1 mM KCl, 0.25 mM KH_2_PO_4_, 1 × 10^−3^ mM H_3_BO_3_, 1 × 10^−3^ mM MnSO_4_, 1 × 10^−4^ mM CuSO_4_, 1 × 10^−3^ mM ZnSO_4_, 5 × 10^−6^ mM (NH_4_)_6_Mo_7_O_24_, and 0.1 mM Fe-EDTA. The plants were grown in a greenhouse of Yangzhou University from July to August in 2019. Three-day-old seedlings were transferred into another nutrient solution with 1 mM Ca(NO_3_)_2_ (control, CK) and 0 mM Ca(NO_3_)_2_ (low N, LN), and the Ca^2+^ concentration in LN was adjusted to the same levels as that of the CK by the addition of CaCl_2_. The experiment was carried out as a completely randomized design with two N levels (CK, 2 mM; LN, 0 mM) [34] and two replicates.

### 2.2. Phenotypic Evaluation and Data Analysis

After twelve days of N treatment, six healthy seedlings were harvested in each replicate per inbred line. The shoots and roots were separated from the first whorl crown roots occur on the stem and stored at 4 °C. A total of 28 traits were evaluated, including 6 shoot traits and 22 root traits (Appendix A). For shoot traits, plant height (PH), leaf length (LL), leaf width (LW) were measured using a ruler, and the SPAD value was measured using a chlorophyll meter SPAD-502PLUS (Konica Minolta, INC., Tokyo, Japan). For root traits, the axial root length of primary root (PRL), seminal root (SRL), and hypocotyl length (HL) were measured using a ruler, and the number of seminal roots (SRN) and crown roots (CRN) was recorded. The SRL divided by SRN was the average length of seminal roots (ASRL). Then, the roots were scanned to produce high resolution images, and the traits total length of primary root (TPRL), primary root surface area (PRSA), average diameter of primary root (APRD), primary root volume (PRV), lateral root length of primary root (PLRL), primary root length between 0 and 0.5 mm in diameter (PRL005), primary root length between 0.5 mm and 1 mm in diameter (PRL0510), primary root length greater than 1.0 mm in diameter (PRL10), total length of seminal root (TSRL), seminal root surface area (SRSA), average diameter of seminal root (ASRD), seminal root volume (TSRV), lateral root length of seminal root (SLRL), seminal root length between 0 and 0.5 mm in diameter (SRL005), seminal root length between 0.5 mm and 1 mm in diameter (SRL0510), and seminal root length greater than 1.0 mm in diameter (SRL10) were analyzed with WinRHIZO Pro 2004b software (Regent Instruments, Sainte Foy, QC, Canada). The roots and shoots were dried at 105 °C for 30 min and then at 55 °C until reaching a constant weight; the root dry weight (RDW) and shoot dry weight (SDW) were recorded. All statistical analyses were performed in R 4.1.0 software package (R Development Core Team 2013, Vienna, Austria). Analysis of variance (ANOVA) implemented in R was used to assess the significance of the differences between treatments, lines, and interactions (G × E). The mean values of each line were used for further analysis. Pearson’s correlation between traits were implemented using R package “psych”. The principal component analysis (PCA) was performed using R packages “FactoMineR” and “factoextra”. The nitrogen response value was obtained according to the value of ((low nitrogen-control)/control). To distinguish the nitrogen response value from the trait *per se*, all acronyms for the nitrogen response value start with the lowercase letter *r*.

### 2.3. Genotypic Data and Genome-Wide Association Analysis

In this study, the genome data of the 213 maize inbred lines was genotyped by genotyping-by-sequencing (GBS). A total of 361,922 SNPs with missing rate ≥ 20% and minor allele frequency ≥ 0.01 of the 213 inbred lines were used for genome-wide association study (GWAS). The GWAS was performed with TASSEL 5.2 software using the mixed linear model (MLM). The first five principal components were applied for covariates to control the effects of population structure (Appendix A). Additionally, a kinship matrix was used to examine the familial relationship across the population. Because of the non-independence of SNPs caused by linkage disequilibrium (LD), it is complicated for the interpretation of statistical significance. A total of 93,457 independent SNPs were determined by the GEC software [35]. The *p* value for determining significant associations was *p* < 1.07 × 10^−5^ (*p* = 1/independent marker number) [33], with a corresponding −log10 (*p*) of 4.97. The population showed rapid LD decay with an average of 50 kb (Appendix A) [33], which was determined with the PopLDdecay software [36]. According to the LD of the association population, all potential genes within 100 kb (50 kb up- and downstream) of the significant loci were identified. The candidate genes functional annotations were obtained from the maizeGDB database (http://www.maizegdb.org, accessed on 20 July 2022). Variant annotations of SNPs were performed by SnpEff [37].

### 2.4. RNA-Seq Analysis of Maize Root under Low N

The maize inbred line B73 was used for the transcriptome analysis. Maize seeds were germinated for 48 h described earlier. Ten germinated seedlings were collected for each biological replicate and immediately frozen in liquid nitrogen. This time point was denoted by S0. Then, the maize seedlings were grown with the paper roll system and transferred to hydroponic solution with the same nutrient composition (CK: 1 mM Ca(NO_3_)_2_; LN: 0 mM Ca(NO_3_)_2_) as described above. The primary roots were harvested for RNA extraction after 1 d, 2 d, and 3 d under low-N, representing S1, S2, and S3. The primary roots of at least six seedlings were pooled to form one biological replicate, and three biological replicates were generated for CK and LN. Total RNA were isolated with the RNeasy Plant Mini kit (Qiagen, Shanghai, China). A total of 1.5 μg RNA per sample was used to construct an RNA-seq library for each sample with the TruSeq RNA Sample Preparation kit (Illumina). All of the 21 libraries were sequenced on an Illumina Hiseq platform and paired-end reads were generated. All raw reads have been deposited into the NCBI Sequence Read Archive (SRA, http://www.ncbi.nlm.nih.gov/sra/, accessed on 20 July 2022) under accession number PRJNA858579. After removing adapters, low-quality reads, and reads containing poly-N sequences from the raw data, the high-quality reads were aligned to the B73 RefGen_V3 reference genome sequence (AGPv3, release 31) using HISAT2 (v2.1.0). The mapped output was processed via FeatureCounts to obtain FPKM (fragments per kilobase of exon model per million mapped reads) value for all genes in each sample [38]. Genes with FPKM > 1 were identified as expressed genes. DEseq 1.8.3 software [39] was used to identify differentially expressed genes (DEGs) between different groups with an adjusted *p* value < 0.05 and foldchange ≥ 2. Gene ontology terms for DEGs were obtained from AgriGO with a corrected *p* value < 0.05.

### 2.5. Quantitative Real-Time PCR

To confirm the RNA-seq results, seven candidate genes were chosen for expression validation using RT-qPCR with gene-specific primers (Appendix A). Root tissues were collected at S0, S1, S2, and S3; three biological replicates were performed for each tissue sample *per* time point. Total RNA was extracted using RNAsimple Total RNA Kit (Tiangen, Beijing, China) and purified using the gDNA wiper Mix (Vazyme, Nanjing, China). cDNA was reverse transcribed using qRT SuperMix II (Vazyme, Nanjing, China). The qRT–PCR reaction was performed in a CFX96 Real-Time System (Bio-Rad, Munich, Germany). The transcript level of each gene was normalized to the expression of the *ZmTubulin1* gene (Zm00001d013367). The relative gene expression was calculated using the 2^−ΔΔ*CT*^ method [40].

### 2.6. Candidate Gene Resequencing and Gene-Based Association Mapping

Genomic DNA was extracted from the fresh young leaves of the 213 maize inbred lines using a standard cetyltrimethyl ammonium bromide (CTAB) protocol. The candidate genes were sequenced by the targeted sequence capture technology of the NimbleGen platform [41] by BGI Life Tech Co., Ltd (Shenzhen, China). The genomic sequences of the candidate genes were mapped onto the B73 reference genome (RefGen_V3). Multiple sequence alignments were analyzed with the MAFFT software (v7.313) [42] and subsequently edited manually. Gene-based polymorphisms [SNPs and insertions and deletions (InDels)] were identified with TASSEL 5.2, with a minor allele frequency (MAF) ≥ 0.05. The significance of the association between maize SNPs and target traits was implemented in TASSEL 5.2 with the mixed linear model. The *p* value threshold to control the genome-wide type 1 error rate was 0.5/n (where n is the number of markers for each candidate gene) PlantPan 3.0 was used for predicting regulatory elements of candidate genes [43].

## 3. Results

### 3.1. Root Traits Differences under Normal and Nitrogen-Deficient Conditions

To determine the root morphological changes induced by LN conditions, a population comprising 213 diverse maize inbred lines was grown under control (CK) and LN conditions in a hydroponic system. A total of 28 traits (22 related to the roots and 6 related to the shoots) were evaluated (Appendix A). There was substantial variability in the observed traits under CK and LN conditions (Figure 1A; Appendix A). Specifically, the coefficient of variation ranged from 11.41% to 58.71% and 14.38% to 66.23% under CK and LN conditions, respectively. A two-way ANOVA of the data for all observed traits revealed that the effects of the genotype, LN treatment, and genotype × environment interaction were significant for most of the investigated traits (*p* < 0.05) (Appendix A). There was a significant correlation between the primary and seminal root traits (Appendix A). Both APRD and SPRD were negatively correlated with root length. The correlation between the primary root traits (PRL, PLRL, TPRL, PRSA, PRL005, and PRL0510) and the seminal root traits (SLRL, ASRL, TSRL, SRL005, and SRL0510) increased in response to N deficiency, which was in contrast to the decrease in the correlation between the root diameter and length in response to the LN supply.

Most of the examined traits were significantly affected by N deficiency, with increases in values ranging from 4% (SLRL) to 50% (SRL). The value of TPRL, PLRL, ASRL, PRL005, PRL, and SRL values increased by more than 30% under LN conditions. In contrast, the LN treatment decreased the LL and SDW values by more than 10% (Figure 1B). Regarding the PCA of the 213 lines under CK and LN conditions, three major principal components accounted for more than 59% of the observed variance (Appendix A). Specifically, PC1 explained more than 36.2% of the variation and was positively associated with the seminar root length (TSRL, SRL00, SLRL, and SRL) and the primary root length (TPRL, PLRL, and PRL005). Additionally, PC2 explained more than 13.4% of the variation and was associated mainly with PRSA, PLRL, TPRL, PRL005, TPRV, and SRN. Moreover, root diameter-related traits, including ASRD and APRD, were associated with PC3, which explained 9.8% of the variation. 

### 3.2. Association Analysis of the Root Traits under Low-N Conditions

A GWAS was conducted for the 28 traits across treatments and the N response value of each trait. A total of 297 significant trait-marker associations were identified for the analyzed traits, including 57, 106, and 98 unique SNPs for the CK condition, LN condition, and N response value, respectively (Figure 2 and Appendix A). The proportion of phenotypic variation explained by each SNP ranged from 10.10% to 17.56%, with a mean of 11.37%. To determine the natural variations in root growth under LN conditions, our subsequent analyses focused on the SNPs detected under LN conditions and the N response value for the root traits. 

Under LN conditions, 106 SNPs were identified as associated with 15 traits. Notably, 15 SNPs on chromosome 2 were associated with 4 primary root traits (PRL005, PLRL, TPRL, and PRSA). The most significantly associated SNP was S2_29754177, which explained 12.7% of the phenotypic variance (Figure 2). One pleiotropic SNP (S10_25265784) was significantly associated with two seminal root traits (SLRL and SRL005) (Appendix A). The number of SNPs associated with each N response value ranged from 1 to 56. Five N response values (rSLRL, rTPRL, rAPRD, rPRL005, and rSRSA) were influenced by a single SNP. A total of 56 SRL0510-related SNPs were mapped of which 23 on chromosome 6 were significantly associated with SRL0510. The most significantly associated SNP was S6_42307320, which explained 11.1% of the phenotypic variance.

### 3.3. Whole-Genome Transcriptome Analysis of the B73 Root in Response to a Low-N Supply

To survey the transcriptomic dynamics of maize root in response to low-N condition, RNA-seq analysis was performed on the B73 root tissues under CK and LN conditions. The RNA-seq yielded 18.9–31.3 million clean reads per sample. The number of expressed genes in all samples varied from 21,647 to 22,878 (Appendix A). Approximately 50% of the expressed gene were in the range 1–10 FPKM, and 40% of the expressed gene were in the range 10–50 FPKM. Approximately 10% of genes exhibited a very high (FPKM ≥ 50) expression level in all samples (Appendix A). 

A total of 1495 differentially expressed genes (DEGs) were identified between CK and LN conditions. There were 852, 463, and 302 genes significantly differentially expressed at the S1, S2, and S3 stages, respectively (Figure 3A). In total, 115 transcription factor (TF)-encoding genes from 28 families were differentially expressed between 2 conditions (Figure 3B). Gene Ontology (GO) term enrichment analyses showed that a variety of different biological processes were affected by nitrate deficit treatment in maize root (Appendix A) of which nine GO terms related to nitrogen metabolism were significantly enriched (FDR < 0.05), including response to nitrate, nitrate transport, nitrate assimilation, nitrate metabolic process, nitrogen cycle metabolic process, reactive nitrogen species metabolic process, nitrogen compound transport, response to organonitrogen compound, and response to nitrogen compound. Likewise, DEGs between CK and LN conditions were also enriched in flavone, flavonoid, and flavanol biosynthetic and metabolic process, organic acid biosynthetic and metabolic process, response to biotic stimulus, response to salicylic acid, and secondary metabolic process (Appendix A). 

Next, we compared the DEGs between different stages under LN conditions. In total, 7084 DEGs were identified in three pairwise comparisons (Figure 3C). Compared with S0 time point, there were 4582 DEGs including 2292 up-regulated genes and 2290 down-regulated genes at S1 time point. Under low N condition, 790 and 2045 genes were significantly up- or down-regulated at the S2 time point compared with S1, respectively. The number of DEGs reduced in the pairwise comparisons of S3LN vs. S2LN (979 up-regulated genes and 745 down-regulated genes). Among these DEGs, 626 TFs were identified and assigned to 45 TF families (Figure 3D). The number of genes in WRKY, bHLH, MYB, ERF, and NAC families exceeded 50. DEGs characterized between different time points of maize root under low-N conditions were enriched in more GO terms, such as response to nitrate, nitrate transport, response to nitrogen compound, nitrogen compound transport, defense response, response to abiotic stimulus, response to acid chemical, response to chemical, response to endogenous stimulus, response to external biotic stimulus, response to hormone, response to organic substance, and response to oxygen-containing compound (Appendix A).

### 3.4. Integrating GWAS and RNA-Seq Data to Prioritize Causal Genes

We used multiple criteria to prioritize the causal genes: (1) location of a SNP iden-tified in the GWAS analysis, (2) annotated genes within a 50-kb window on each side of the SNP (100-kb window total), (3) relative expression of the 100-kb window genes between treatments or stages under LN conditions, and/or presence of amino acid polymorphism in CDS (coding DNA sequence) region of candidate gene, and (5) possible gene function. On the basis of the whole-genome LD decay, a total of 257 candidate genes were detected within the 50-kb window of identified SNPs, including 185 genes associated with LN conditions or the N response value for root traits (Appendix A). Among these genes, 48 were significantly differentially expressed between treatments or stages under LN condition (Figure 4, Appendix A). The functional variance in the coding sequence act on target traits because of allelic nt/aa differences impacting protein function. Variation in CDS leads to amino acid polymorphism of four candidate genes (GRMZM2G024054, GRMZM2G449274, GRMZM2G361362, and GRMZM2G404922) (Appendix A). The expression levels of GRMZM2G404922 were significantly different between S2 and S3 under a low-N condition. Additionally, RT-qPCR experiments were used to validated the expression pattern of seven candidate genes. There was strong positive correlation (*p* < 0.05) of each gene between the RNA-seq and RT-qPCR. The Pearson’s correlation coefficient (*r*) ranged from 0.833 to 0.995. These results indicated the reliability of our transcriptomic profiling data (Appendix A).

The loci associated with maize traits under CK conditions included 13 candidate genes, encoding transcription factors and other proteins. For example, GRMZM2G159500, which encodes the NAC (NAM-ATAF-CUC2) 29 transcription factor, was identified as the candidate causal gene for S9_133064384, which was associated with APRD_CK. Under LN conditions, 18 candidate causal genes were identified. These genes mainly encode transcription factors, enzymes, and other proteins with unknown functions. The cytokinin dehydrogenase gene *ZmCKO2* (GRMZM2G050997), which contributes to zeatin biosynthesis, was identified as the candidate gene for S3_152244542 and S3_152244553, which were associated with SRL005_LN. The methionine synthase 1 gene GRMZM2G149751 was revealed as the candidate causal gene for four pleiotropic SNPs (S1_176870882, S1_176871410, S1_176872498, and S1_176872647), which were associated with ASRL. One candidate gene (GRMZM2G081930) encoding a NAC (NAM/ATAF1/2/CUC2) transcription factor was detected in the 29,705,368–29,754,177 base pair (bp) region on chromosome 2. This region comprised 15 SNPs significantly associated with TPRL, PLRL, PRL005, and/or PRSA. The candidate gene GRMZM5G881641 was simultaneously associated with ASRD under LN conditions and the relative value. Sixteen candidate genes were detected for the N response value (Appendix A).

### 3.5. Natural Variation of ZmNAC36 Affect Root Traits in Response to Nitrogen Deficiency

The GWAS results indicated that 15 SNPs on chromosome 2 were significantly associated with 4 primary root traits (TPRL, PLRL, PRL005, and PRSA) under LN conditions (Figure 5A). Two genes (GRMZM2G081930 and GRMZM2G380138) were detected in the LD region (Figure 5A) of which GRMZM2G380138 was not differentially expressed between treatments and stages under LN conditions. The GRMZM2G081930 expression level gradually increased during the root development period under both CK and LN conditions. In the S3 time point, the GRMZM2G081930 expression level was significantly higher under the LN condition than under the CK condition (Figure 5B). Accordingly, *ZmNAC36* (GRMZM2G081930) is a candidate gene that warrants further study. To investigate the association between the allelic and root variations in the association panel, we resequenced *ZmNAC36* in 213 maize inbred lines. We analyzed the *ZmNAC36* genomic region comprising 5657 bp, including the 1621-bp upstream region, the 3478-bp coding region, and the 558-bp downstream region. A total of 591 sequence variations were identified (MAF ≥ 0.05), including 408 SNPs and 183 InDels. A marker–trait association analysis conducted on the basis of the MLM identified 38 SNPs and 12 InDels that were significantly associated with TPRL_LN, PLRL_LN, PRL005_LN, and PRSA_LN (Figure 6A). These variations could be divided into two blocks (Figure 6B and Appendix A). The first block contained 26 SNPs and 7 InDels located in the promoter region of *ZmNAC36*. These variations explained 10.30%, 11.03%, 10.11%, and 9.82% of the phenotypic variation for TPRL, PLRL, PRL005, and PRSA. The second block contained 12 SNPs and 5 InDels located in the intron region. Two major haplotypes emerged from these variations across 213 inbred lines. A significant phenotypic difference was revealed via an ANOVA of the haplotypes for TPRL, PLRL, PRL005, and PRSA under LN conditions (Figure 6C).

## 4. Discussion

### 4.1. Effects of Low-N Conditions on the Root Architecture

Nitrogen is one of the most important input factors influencing crop production. Previous studies demonstrated that breeding N-efficient cultivars is an effective strategy for decreasing the environmental pollution caused by excessive N fertilizer applications [44,45]. The root system is the primary plant organ for absorbing nutrients and water and responding to various environmental stresses. However, it was difficult to visualize and measure root structures and their growth compared to aboveground structures. Previous studies have indicated that the hydroponic cultivation system can be used for high-throughput phenotypic screens of maize at the early stages [7,13,46]. The plasticity of the root system architecture is at least partly due to the spatiotemporal heterogeneity of soil [47]. Numerous studies suggested that nitrate has critical effects on root growth, development, and architecture [48,49,50]. In an earlier study on Arabidopsis, extensive natural variations were observed in the primary root length [27], total lateral root length, and total root length among 200 accessions [14], with longer roots under LN conditions than under HN conditions. In another study, which involved rice, plants were significantly shorter under LN conditions than under normal conditions [51]. In contrast, an LN supply reportedly promotes primary root growth and lateral root elongation [52]. In maize, compared with HN conditions, the Ye478 and Wu312 primary root and seminal root lengths increase in response to LN stress, but the root systems of these two inbred lines are differentially sensitive to the N treatment [7]. Sun et al. examined the root traits of 461 maize inbred lines under HN and LN conditions, and detected increases in the total root length, axial root length, and lateral root length under LN conditions [46]. In the current study, the root and shoot traits varied considerably among the 213 maize inbred lines under CK and LN conditions, with a high level of plasticity in response to the LN treatment. The values for the root traits (TPRL, PLRL, ASRL, PRL005, PRL, and SRL) increased under LN conditions, whereas the values for the shoot traits (PH, LL, LW, and SPAD) were lower under LN conditions than under CK conditions. In accordance with previous studies [12,46,53], these results reflected the strong genotype × N level interactions affecting root traits. Moreover, under LN conditions, the maize root architecture was modified to mine the substrate for more N.

### 4.2. Candidate Genes Identified in the Detected QTL Regions for Root Traits under Low-N Conditions

Root characteristics are inherently complex traits. In previous studies, numerous QTLs associated with root traits at different nitrate levels were identified in multiple maize populations via a QTL analysis [7,44] and GWAS [27,46]. Sun et al. [46] conducted a GWAS to elucidate the natural variations in root traits under LN conditions, and obtained 328 significant SNPs associated with 21 root traits and three shoot traits during the seedling emergence stage under HN and LN conditions. Adopting the same plant cultivation method, we identified 297 significant SNP–trait associations for different N levels and the N response value via a GWAS. Unlike earlier studies, there were no overlapping loci among populations, indicative of the considerable genetic variation and complex genetic mechanism underlying root trait changes induced by nitrate deficiency. 

Because LD often involves a relatively broad region surrounding each significant locus, candidate genes often remain unknown. Several previous studies suggested that integrating GWAS results and gene expression data is a powerful way to detect candidate genes [33,54,55,56]. In the current study, the integration of GWAS and DEG data led to the identification of 48 candidate genes. These candidate genes encode enzyme catalyzing reactions in hormone biosynthesis pathways as well as transcription factors and other diverse proteins. A cytokinin oxidase 2 gene (*ZmCKO2*) belonging to the cytokinin oxidase/dehydrogenase (CKX) gene family was associated with SRL005_LN. Cytokinin is a well-known inhibitor of root elongation [57,58]. The expression of *CKX* genes clearly affects root growth in Arabidopsis [57], rice [59,60], wheat [61], barley [62], and maize [63]. Waidmann et al. conducted a GWAS, which revealed that the inhibitory effect of the cytokinin signaling pathway on Arabidopsis lateral root growth involves *AtCKX2* [64]. Wang et al. identified 12 *ZmCKX5* variants that are significantly associated with six root traits, while also demonstrating that *ZmCKX5* may regulate maize root development [65]. Gao et al. showed that *OsCKX4* regulates root growth via auxin and cytokinin pathways [59]. These results suggest that the mechanism underlying the regulatory effects of *CKXs* on root growth may vary among plant species, but this will need to be experimentally confirmed. In the current study, we detected two NAC transcription factor genes, *ZmNAC29* and *ZmNAC36*, with the former associated with APRD_CK and the latter associated with TPRL_LN, PLRL_LN, PRL005_LN, and PRSA_LN. Mao et al. concluded that *OsNAC2* negatively regulates rice root development by decreasing the number of crown roots and the root length as well as by binding directly to the *OsCKX4* promoter to modulate expression [66]. Tang et al. identified the NUE-related transcription factor OsNAC42, which binds directly to the promoter of the NUE-related gene *OsNPF6.1* to activate expression; the rare natural allele *OsNPF6.1^HapB^* enhances nitrate uptake and confers a high NUE by increasing the yield under LN conditions [67]. Xu et al. identified a nitrate-inducible transcription factor (NAC056) that regulates the expression of genes mediating nitrate assimilation and that promotes lateral root growth in Arabidopsis [68]. Vidal et al. demonstrated that the NAC4 transcription factor functions downstream of the auxin receptor AFB3 to control lateral root growth in response to nitrate [69]. He et al. isolated the nitrate-inducible transcription factor TaNAC2-5A from wheat and suggested that the overexpression of *TaNAC2-5A* enhances root growth and increases the ability of roots to acquire N [70]. These results indicate that *ZmNAC36* may regulate root growth and responses to LN conditions. 

In this study, a candidate gene association analysis was performed to identify the SNPs and InDels in the *ZmNAC36* genic region among maize inbred lines. We detected 38 SNPs and 12 InDels significantly associated with TPRL, PLRL, PRL005, and PRSA under LN conditions of which 26 SNPs and 7 InDels were in the *ZmNAC36* promoter region. These variants explained 10.30%, 11.03%, 10.11%, and 9.82% of the variation in TPRL, PLRL, PRL005, and PRSA, respectively. Previous studies confirmed that sequence variations in the promoter region may be related to gene expression changes affecting certain phenotypes. In maize, genetic variations in the promoter region of *ZmTIP1* were found to be significantly associated with the survival rate of maize seedlings under drought stress. Elevated expression due to sequence differences in the *ZmTIP1* promoter but not the protein-coding region are responsible for the gene functional variation in drought tolerance [71]. An 82-bp miniature inverted-repeat transposable element (MITE) insertion in the *ZmNAC111* promoter repressed *ZmNAC111* expression via RNA-directed DNA methylation and H3K9 demethylation, and it was significantly associated with drought tolerance [72]. In wheat, a 159-bp MITE insertion in *TaMOR-B* promoter caused DNA methylation and a lower expression of *TaMOR-B*, and associated with lower root dry weight and shorter plant height [73]. A 108-bp insertion in the promoter of *TaNAC071-A* in wheat leads to increased expression and enhanced drought tolerance, and the insertion allele is directly activated by TaMYBL1, thereby leading to increased *TaNAC071-A* expression and drought tolerance [74]. Additionally, sequence variations in the promoter regions of *ZmVPP1* [75], *ZmPP2C-A10* [76], *bsr-d1* [77], and *SlALMT9* [78] are associated with changes to agronomic traits. Here, we observed that variants in the promoter region of *ZmNAC36* located in *cis*-regulatory regions, which may lead to the changes in the promoter binding sites (Appendix A). The roles of *ZmNAC36* during the maize response to nitrate deficiency will need to be further analyzed. Our findings provide an important foundation for future investigations of the molecular basis of root responses to LN conditions and for exploiting specific genes to modulate root traits to increase the NUE in maize.

## 5. Conclusions

In summary, a genome-wide association study was conducted to analyze root traits and their response to N deficiency. We detected 297 significant SNP of which 57, 106, and 98 were associated with the control condition, LN condition, and N response value, respectively. A total of 51 candidate genes identified by genome-wide association study were supported by the differentially expressed genes or SNP in CDS region that cause amino acid alteration. A candidate gene *ZmNAC36* was resequenced in all tested lines, and 33 variants in the promoter region were significantly associated with 4 root traits under the LN condition. The molecular mechanism of natural variation of Zm*NAC36* contributes to root development under low N needs to be further studied. Collectively, our findings provide potential loci and genes useful for the genetic improvement of root traits under low nitrogen condition.

## Figures and Tables

**Figure 1 genes-13-01632-f001:**
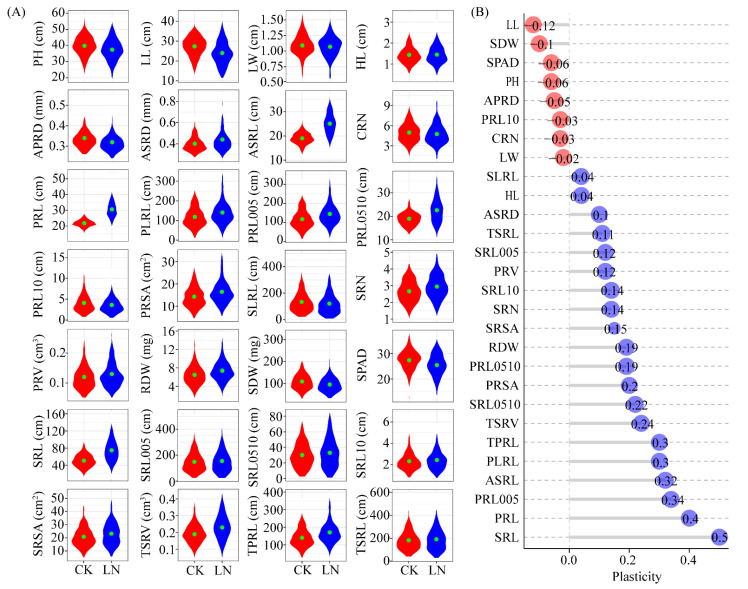
Natural variation of root plasticity in response to nitrate deficiency. (**A**) Phenotypic variation in plant height (PH), leaf length (LL), leaf width (LW), hypocotyl length (HL), average diameter of primary root (APRD), average diameter of seminal root (ASRD), average length of seminal roots (ASRL), crown root number (CRN), primary root length (PRL), lateral root length of primary root (PLRL), primary root length between 0 and 0.5 mm in diameter (PRL005), primary root length between 0.5 mm and 1 mm in diameter (PRL0510), primary root length greater than 1.0 mm in diameter (PRL10), primary root surface area (PRSA), lateral root length of seminal root (SLRL), seminal roots number (SRN), primary root volume (PRV), root dry weight (RDW), shoot dry weight (SDW), SPAD, seminal root length (SRL), seminal root length between 0 and 0.5 mm in diameter (SRL005), seminal root length between 0.5 mm and 1 mm in diameter (SRL0510), seminal root length greater than 1.0 mm in diameter (SRL10), seminal root surface area (SRSA), total seminal roots volume (TSRV), total length of primary root (TPRL) and total length of seminal root (TSRL) were presented with violin plots of 213 maize inbred lines under CK (red) and LN (blue) conditions, respectively. (**B**) Phenotypic plasticity observed of 28 traits in 213 maize inbred lines. The phenotypic plasticity = (low nitrogen-control)/control.

**Figure 2 genes-13-01632-f002:**
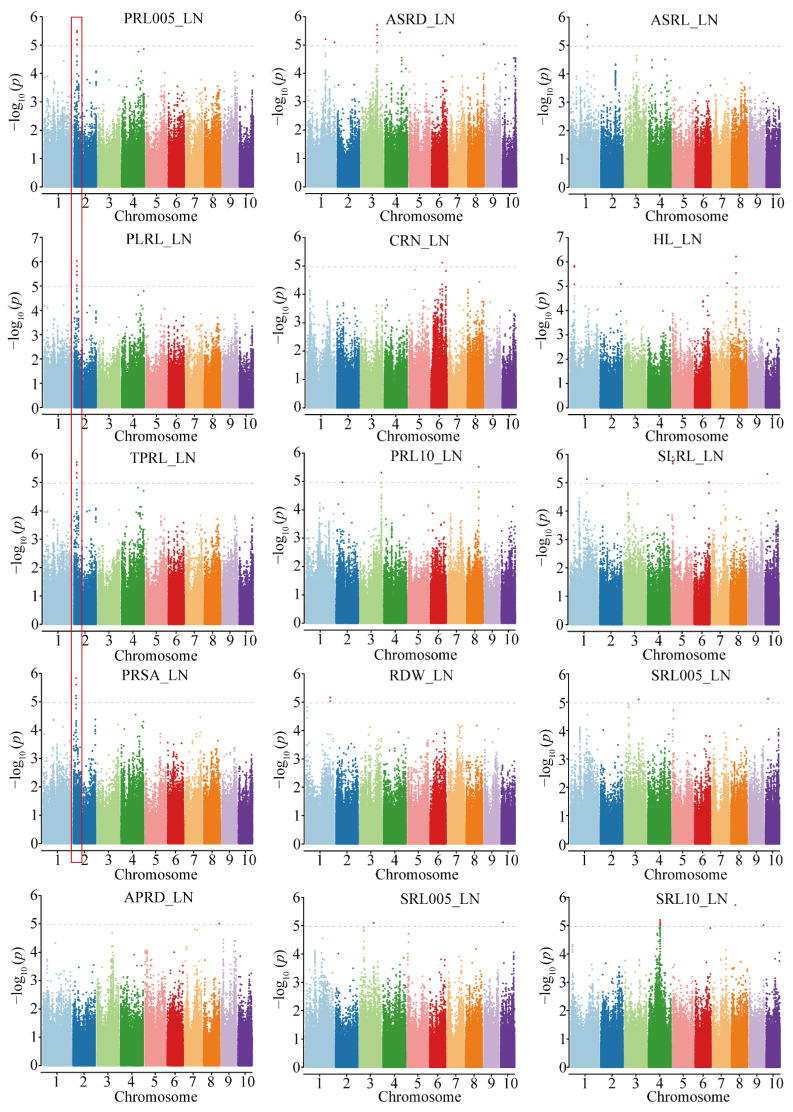
Manhattan plots of associated traits under low-N condition.

**Figure 3 genes-13-01632-f003:**
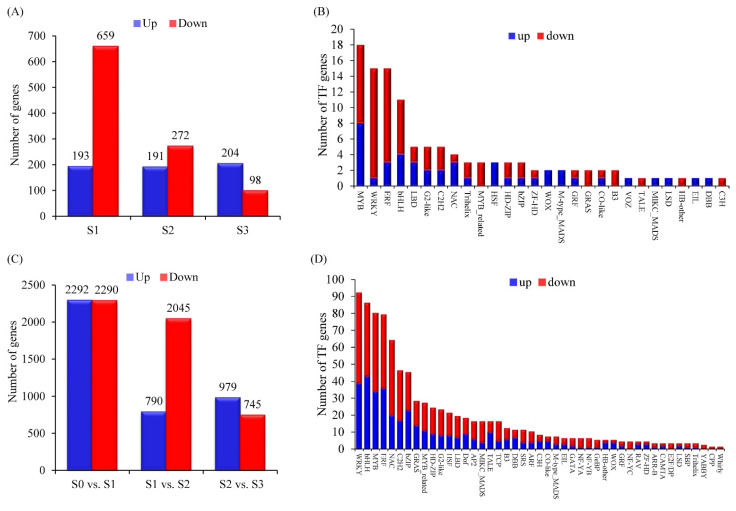
Overview of differentially expressed genes (DEGs). Numbers of different expressed genes (**A**) and transcription factors (**B**) (up-regulated genes highlighted in blue box and down-regulated genes highlighted in red box) between control and low nitrate conditions. Numbers of DEGs (**C**) and transcription factors (**D**) during root development under low nitrate condition.

**Figure 4 genes-13-01632-f004:**
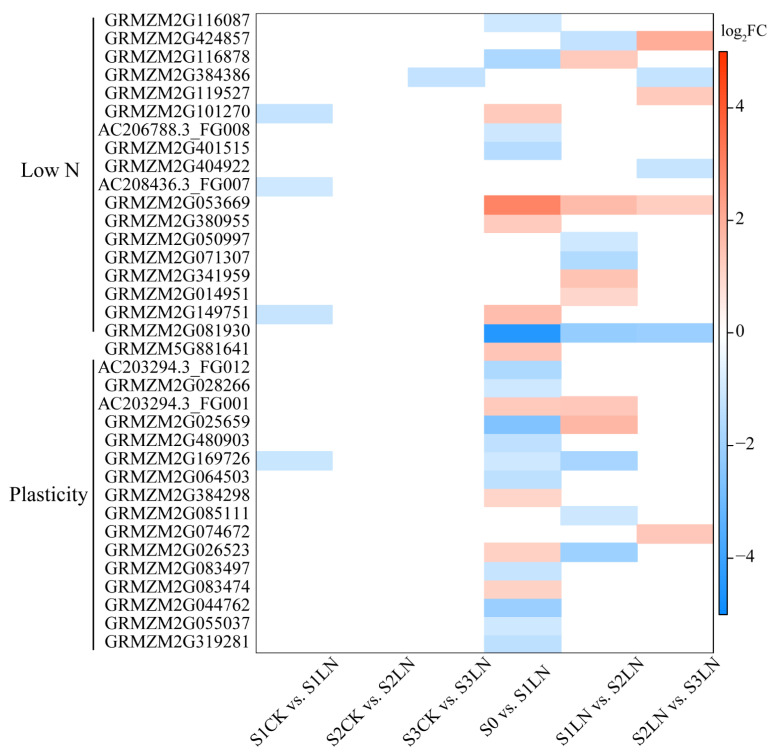
Candidate genes identified by integrating GWAS and RNA-seq. The heatmap plot showed the prioritized causal genes identified under low N condition and/or root plasticity. The log2 (foldchange) was shown by the shade of colors.

**Figure 5 genes-13-01632-f005:**
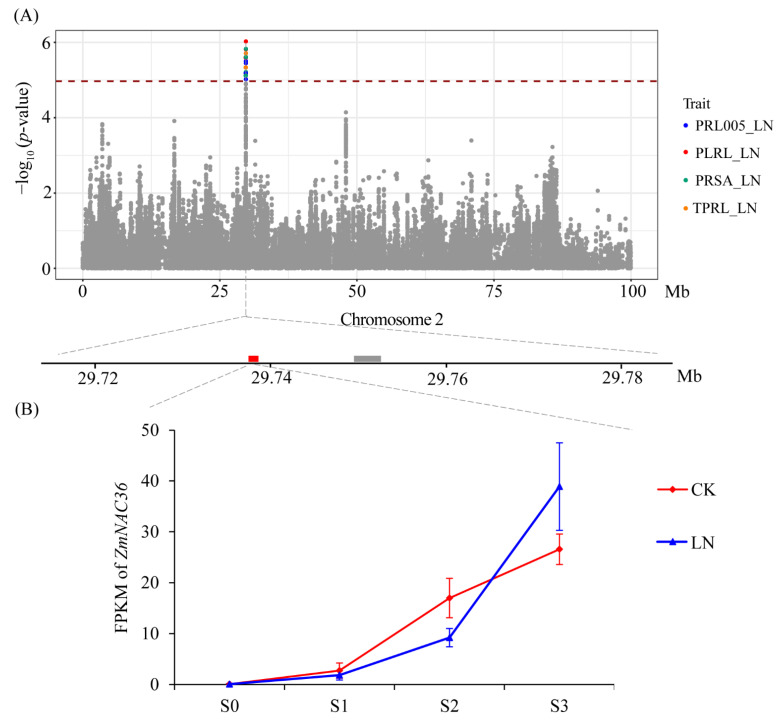
Association analysis of pleiotropic SNPs on chromosome 2 with PRL005_LN, PLRL_LN, TPRL_LN, and PRSA_LN. (**A**) Manhattan plot for PRL005, PLRL, TPRL, and PRSA under low N condition. The lower plot showed genes surrounding SNPs on chromosome 2 which were associated with PRL005_LN, PLRL_LN, TPRL_LN, and PRSA_LN. (**B**) Expression levels of *ZmNAC36* in B73 root at different time point under control and low N conditions.

**Figure 6 genes-13-01632-f006:**
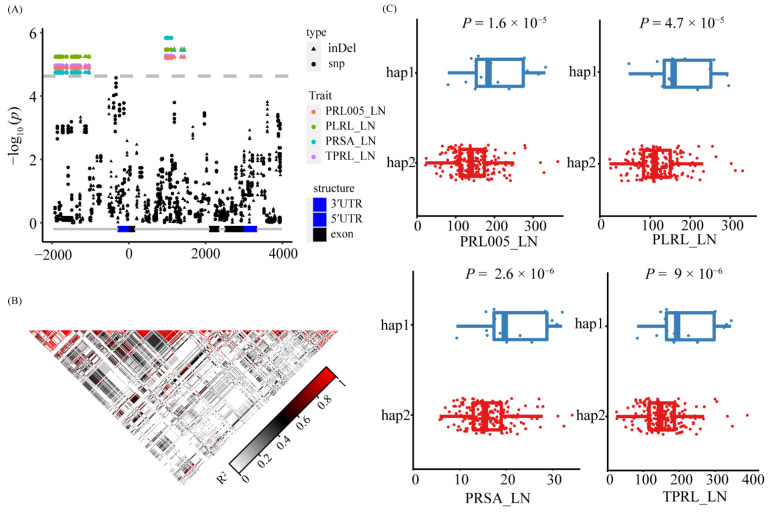
Candidate gene association analysis and haplotype analysis of *ZmNAC36*. (**A**) Local Manhattan plot within *ZmNAC36*. The InDels are shaped by triangle and SNPs are shaped by dot. (**B**) The pattern of the pairwise linkage disequilibrium (LD) of the genetic variants. (**C**) Phenotypic differences between different haplotypes. The statistical significances were determined by a two-sided Student’s *t*-test.

## Data Availability

The data sets supporting the results of this article are included within the article (and its Appendix A). Raw reads of all samples have been deposited in the NCBI Sequence Read Archive (SRA, http:www.ncbi.nlm.nih.gov/sra/) under accession number PRJNA858579.

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
