# Peer review of "GWAS and Transcriptome Analysis Reveal Key Genes Affecting Root Growth under Low Nitrogen Supply in Maize"

_genes, 2022, doi:10.3390/genes13091632_

Round 1
Reviewer 1 Report
Comments on Genes-1865949
I have the following comments on this manuscript.
1. Need to re-check the spell mistake e.g., Figure 6A “Chromosome 2” and sentence structure e.g., line # 138-139 “analysis of variance (ANOVA) as implemented….” throughout the research paper.
2. In the abstract avoid using the abbreviations, at first moment must write full name of the trait, otherwise it creates difficulty for readers understanding.
3. In the methodology section authors mentioned the 28 traits but did not enlist the name of all traits. Please enlist all traits with abbreviations to avoid confusion and misleading information.
4. What is the basis to use first five principal components as covariate? Please explain?
5. Author used threshold value to select significant SNPs using formula (P = 1/independent marker number), instead of commonly used Bonferroni correction method… why? Provide the reference for this formula.
6. Author mentioned the LD decay in the methodology section but in the result section no figure/results showed. Kindly provide the LD decay graph from which author estimate the LD decay.
7. In the result section, author should provide the violin plots for all traits instead of selected four traits.
8. Figure 1B, plasticity values inside the circles are not visible, make it clear using lighter colors.
9. In the table S2, mean square value of each trait ANOVA also required with the significance asterisk.
10. Author provided the four trait’s Manhattan plots under low -N condition. But author did not provide the Manhattan plots for the rest of 24 traits under low -N and control condition. Did author check the 2A associated SNPs are specific to low nitrogen condition? Please elaborate the GWAS results and showed the Manhattan plot of all traits under control condition in supplementary figures.
11. Correct the “3.3 Global transcriptome analysis….” to “3.3 whole-genome transcriptome analysis…”
12. Results for “Integrating GWAS and RNA-seq data to prioritize causal genes” is not properly written. Please re-write these results and associate the expression of each gene with the GWAS signals and explain who author prioritize the causal genes. And then explain the function of col-localized gene. As author identify the causal gene for four traits GRMZM2G081930, and also show higher expression pattern under LN in figure 6B. But the expression heatmap results in Figure 5 seems different. Please explain this.
13. I am wondering how ZmNAC36 (GRMZM2G081930) is candidate gene. Either author did not identify any SNP variation in the exon region. Can author try to explain the mechanism and provide the clue how promoter variations control the nitrogen uptake under N-deficit condition?
14. Authors used different abbreviations. I will suggest them to use the complete form at first appearance and then use abbreviations.
15. Authors used LN and HN. On which bases are they categorized? please provide the N concentrations for LN and HN.
16. How many seedlings were transferred in each treatment and replication?
17. In the abstract authors used accessions and here they used inbred lines. technically both are different, I will suggest them to clarify it and use the correct form in the whole manuscript.
18. Why the authors used B73? On which characteristics did they choose B73 for RNA-seq?
19. Authors find different SNPs. how do they confirm the false positive results?
20. For better understanding, I will suggest to the authors provide the full names of the parameters in the main text.
21. Authors found 48 candidate genes through the RNA-seq approach, but they did not validate the RNA-Seq Data by different approaches such as qRT-PCR. This is very important for this study.
22. Rewrite the conclusion part and provide some strong future recommendations.
23. Check the references carefully and remove the old ones and add the latest.
24. Gene names and species names must be italic. check the whole manuscript.

Author Response
Dear reviewer,
Thanks a lot for your consideration of our manuscript for publication in Genes pending a major revision. In the revised version we had carefully considered all of the points you had raised. Our responses to the comments point to point are presented as below, and the corresponding changes are in the text of revised manuscript.
Thanks again for your consideration of our manuscript. We would be glad to respond to any further questions and comments that you may have.
Yours sincerely,
Pengcheng Li
-----------------------------------------------------------------------------------------------------------------
In response to the reviewer 1 comments: (Reviewer’s comments are marked in blod)
- Need to re-check the spell mistake e.g., Figure 6A “Chromosome 2” and sentence structure e.g., line # 138-139 “analysis of variance (ANOVA) as implemented….” throughout the research paper.
R: Thank you for your comments. We have checked and corrected the spell mistake in the revised manuscript.
- In the abstract avoid using the abbreviations, at first moment must write full name of the trait, otherwise it creates difficulty for readers understanding.
R: Thank you very much. We have revised the abbreviations in the abstract as you suggested. We have written the full name of the traits in the abstract.
- In the methodology section authors mentioned the 28 traits but did not enlist the name of all traits. Please enlist all traits with abbreviations to avoid confusion and misleading information.
R: Thank you very much for this comment. We had accepted and enlisted the name of all traits in the methodology section (Line 140-155).
- What is the basis to use first five principal components as covariate? Please explain?
R: Thank you very much for this comment. The first five principal components cumulatively explained 22.08% of the variance, and the first two PCs do a good job of distinguishing different inbred lines (Li et al., 2021, Theoretical and Applied Genetics). And we have added the plot in Supplementary Figure S1 to show the effect of each principal component.
- Author used threshold value to select significant SNPs using formula (P = 1/independent marker number), instead of commonly used Bonferroni correction method… why? Provide the reference for this formula.
R: Thanks for your questions. The Bonferroni correction resets the significance threshold from α to α/M in the presence of M independent tests, it is probably the commonly method for multiple testing adjustment. However, the Bonferroni correction assumes independence among the test considered, so that it is inherently conservative when considering SNPs in LD block. The testing of a huge number of SNPs needs to be taken into account in the interpretation of statistical significance in such genome-wide studies, Li et al. (2012, Human Genetics) proposed the use of the effective number of independent markers (Me) for adjustment of multiple testing. Additionally, the corresponding reference was added in the References section.
- Author mentioned the LD decay in the methodology section but in the result section no figure/results showed. Kindly provide the LD decay graph from which author estimate the LD decay.
R: Thank you a lot for this comment. The linkage disequilibrium (LD) decay (r2 < 0.2) in the population is approximately 50 kb (Li et al., 2021, Theoretical and Applied Genetics). We have added the LD decay graph in the Supplementary Figure S2.
- In the result section, author should provide the violin plots for all traits instead of selected four traits.
R: Thank you for your review. We have provided the violin plots for all traits in Figure 1A.
- Figure 1B, plasticity values inside the circles are not visible, make it clear using lighter colors.
R: Thanks for this comment. We have modified the colors of the circles in Figure 1B to make it clearer.
- In the table S2, mean square value of each trait ANOVA also required with the significance asterisk.
R: Thanks very much for pointing out this. We have added the mean square value of each trait in the Supplementary Table S2.
- Author provided the four trait’s Manhattan plots under low -N condition. But author did not provide the Manhattan plots for the rest of 24 traits under low -N and control condition. Did author check the 2A associated SNPs are specific to low nitrogen condition? Please elaborate the GWAS results and showed the Manhattan plot of all traits under control condition in supplementary figures.
R: Thank you very much for this comment. The associated SNPs in Figure 2 were not specific to low-N condition, but the association networks between SNPs and traits under low-N, control conditions, and the trait plasticity. We plotted Manhattan plot of all traits under two treatments and the N response value of each trait. The current Figure 2 was moved to Supplementary Figure S3. And Manhattan plots of traits which were significantly associated with corresponding SNPs under low-N condition was used as Figure 2 in the revised manuscript. In addition, we added the Manhattan plots of traits under control condition and the N response value in Supplementary Figure S4 and S5, respectively.
- Correct the “3.3 Global transcriptome analysis….” to “3.3 whole-genome transcriptome analysis…”.
R: Thanks for this comment. We have corrected it in the revised manuscript.
- Results for “Integrating GWAS and RNA-seq data to prioritize causal genes” is not properly written. Please re-write these results and associate the expression of each gene with the GWAS signals and explain who author prioritize the causal genes. And then explain the function of col-localized gene. As author identify the causal gene for four traits GRMZM2G081930, and also show higher expression pattern under LN in figure 6B. But the expression heatmap results in Figure 5 seems different. Please explain this.
R: Thank you very much. The candidate genes did not always manifest by a difference in gene expression, they act on target traits because of allelic differences impacting protein function. We rephrased these results and provided the criteria to prioritize the causal genes as follows: (1) location of a SNP identified in the GWAS analysis, (2) annotated genes within a 50-kb window on each side of the SNP (100-kb window total), (3) relative expression of the 100-kb window genes between treatments or stages under LN condition, and/or presence of amino acid polymorphism in CDS (coding DNA sequence) region of candidate gene, and (5) possible gene function. Please see Lines 419-424 in our revised manuscript.
In Figure 5, the heatmap showed the candidate genes which were differentially expressed between under different nitrogen levels or durations of nitrogen deficiency, rather than the expression levels of genes. The log2(foldchange) was shown by the shade of colors.
- I am wondering how ZmNAC36 (GRMZM2G081930) is candidate gene. Either author did not identify any SNP variation in the exon region. Can author try to explain the mechanism and provide the clue how promoter variations control the nitrogen uptake under N-deficit condition?
R: Thank you for your comment. Several studies have pointed out that sequence variations in the promoter region will directly affect gene expression level and ultimately lead to phenotypic differences (Mao et al., 2015, Nat Commun; Mao et al., 2022, Mol Plant; Wang et al., 2016, Nat Genet; Xiang et al., 2017, Mol Plant; Li et al., 2017, Cell; Ye et al., 2017, Plant Cell). A 108-bp insertion in the promoter of TaNAC071-A in wheat leads to increased expression and enhanced drought tolerance, and the insertion allele is directly activated by TaMYBL1, thereby leading to increased TaNAC071-A expression and drought tolerance. Genetic variations in the promoter region of ZmTIP1 were found to be significantly associated with the survival rate (SR) of maize seedlings under drought stress. Elevated expression due to sequence differences in the ZmTIP1 promoter but not the protein-coding region are responsible for the gene functional variation in drought tolerance (Zhang et al., 2021, Plant Biotechnology Journal). Here, we observed that variants in the promoter region of ZmNAC36 located in cis-regulatory regions, which may be lead to the changes in the promoter binding sites (Figure S11, Table S10). The roles of ZmNAC36 during the maize response to nitrate deficiency will need to be further analyzed. We discussed this point in the revised manuscript (Line 610-612).
- Authors used different abbreviations. I will suggest them to use the complete form at first appearance and then use abbreviations.
R: Thanks for your suggestion. We have checked and revised seriously following your comments.
- Authors used LN and HN. On which bases are they categorized? Please provide the N concentrations for LN and HN.
R: Thank you for your review. The two different nutrient solutions were control (CK, 2 mM) and low-N (LN, 0 mM). We added the N concentrations for LN and CK in the methodology section of revised manuscript (Line 135).
- How many seedlings were transferred in each treatment and replication?
R: Thank you. Two paper rolls were selected as experimental replicates for each maize inbred line in each treatment (Line 134-135). Eight plants were taken as a replicate (Line 122).
- In the abstract authors used accessions and here they used inbred lines. technically both are different, I will suggest them to clarify it and use the correct form in the whole manuscript.
R: Thanks for pointing out this. The panel used in this study comprised 213 inbred lines collected from tropical, subtropical or temperate zone (Xu et al., 2022, The Crop Journal; Li et al., 2018, Planta). We have checked and revised in the whole manuscript.
- Why the authors used B73? On which characteristics did they choose B73 for RNA-seq?
R: Thank you for this comment. B73 is a reference genome and has long been a major resource for genetics and molecular biology research. Hirsch et al. found that alignment rate for RNA-seq data from non-B73 genotypes to the B73 reference genome is approximately 13% lower than the alignment rate of RNA-seq data generated from B73 plants (Hirsch et al., 2014, The Plant Cell).
- Authors find different SNPs. how do they confirm the false positive results?
R: Thank you for your comment. First, a modified Bonferroni correction was used to select a threshold P-value to identify significant SNPs (Li et al., 2012, Human Genetics; Li et al., 2018, Frontiers in Plant Science). Second, gene expression is informative and easily measurable source of functional information. Several previous studies have used GWAS and RNA-seq to characterize GWAS results in maize (Schaeferet al., 2018, The Plant Cell; Sekhon al., 2014, Plant Physiology). In our study, RNA-seq analysis was used as one of the criteria to prioritize the causal genes. Additionally, we generated ZmNAC36 transgenic maize plants for further functional analysis.
- For better understanding, I will suggest to the authors provide the full names of the parameters in the main text.
R: Thank you very much for this suggestion. We have provided the full names of traits in the main text.
- Authors found 48 candidate genes through the RNA-seq approach, but they did not validate the RNA-Seq Data by different approaches such as qRT-PCR. This is very important for this study.
R: The results of RNA-seq were validated via RT-qPCR. There was strong positive correlation (P < 0.05) of each gene between the RNA-seq and RT-qPCR. The Pearson’s correlation coefficient (r) ranged from 0.833 to 0.995. These results indicated the reliability of our transcriptomic profiling data. The method and results of RT-qPCR were provided in Line 215-225, Line 433-437 and Figure S9 of the revised manuscript, respectively.
- Rewrite the conclusion part and provide some strong future recommendations.
R: Thank you. Your question made us think more about our results. The conclusion part had been re-written and revised seriously following your comments.
- Check the references carefully and remove the old ones and add the latest.
R: Thank you for your comments. We have checked and revised the references in the revised manuscript.
- Gene names and species names must be italic. check the whole manuscript.
R: Thanks for your review. We have carefully checked and revised the full text.
Reviewer 2 Report
I have found the ms by Wang et al rather well written and reporting interesting data in a very important area, namely the root response to differential N fertilization.
I have a few relatively minor comments, which however, in my opinion, should be addressed.
1. General comment. The authors should be aware and should tell the reader that not always a candidate gene manifests by a difference in gene expression. Candidate genes may act on target traits because of allelic nt/aa differences impacting protein function. In this case, there is no need for differences in allele expression levels. The authors are invited to consider this when they explain their strategy and their results.
Line 29. I would rather say ‘to improve our understanding’
line 109. Type of plant materials is a key information and should not be left to references. PLease provide at least the main information about this collection
Line 202. Not sure whether 'natural population' is the right definition here. Indeed this is impossible to comment without the information about the collection as I said in the above comment.
Line 230, caption figure 1. Units are missing in the X and Y axis of 1A and 1B graphs. Specifically PRL should probably be PRL (cm), etc. PLease use unit for Plasticity too in 1B, or please provide an explanation about Plasticity in the caption
Line 239. Apparently the authors did not plot the GWAS results for the control conditions. Why? I would welcome those graphs in the main text. On the opposite, I do not see much information in the current fig 2, I suggest to move it in supplementary.
DISCUSSION. The authors must comment on the fact that the root developmental stage they have targeted is very early in maize development; it is actually a time window when, in my opinion, the heterotrophic phase (eg. surviving of the seedling based on seed resources) is not even completed.
Line 414. I would say 'often involves'
Author Response
Dear reviewer,
Thanks a lot for your consideration of our manuscript for publication in Genes pending a major revision. In the revised version we had carefully considered all of the points you had raised. Our responses to the comments point to point are presented as below, and the corresponding changes are in the text of revised manuscript.
Thanks again for your consideration of our manuscript. We would be glad to respond to any further questions and comments that you may have.
Yours sincerely,
Pengcheng Li
In response to the reviewer 2 comments: (Reviewer’s comments are marked in bold)
- General comment. The authors should be aware and should tell the reader that not always a candidate gene manifests by a difference in gene expression. Candidate genes may act on target traits because of allelic nt/aa differences impacting protein function. In this case, there is no need for differences in allele expression levels. The authors are invited to consider this when they explain their strategy and their results.
R: We appreciate very much for your valuable comments and constructive suggestions. We further analyzed the sequence variations in the coding region and finally identified four candidate genes with variations in CDS leads to amino acid polymorphism (GRMZM2G024054, GRMZM2G449274, GRMZM2G361362, and GRMZM2G404922) (Line 429-431).
- Line 29. I would rather say ‘to improve our understanding’
R: Thank you very much for this suggestion.
- Line 109. Type of plant materials is a key information and should not be left to references. Please provide at least the main information about this collection
R: Thank you very much for this comment. The panel used in this study comprised 213 maize inbred lines were collected from a wide range of geographical locations, including five heterotic groups (Reid, Lancaster, Tang Si Ping Tou, Zi330, and mixed group) in China (Xu et al., 2022, The Crop Journal; Li et al., 2018, Planta). We have added the information in the revised manuscript in Line 117-118.
- Line 202. Not sure whether 'natural population' is the right definition here. Indeed this is impossible to comment without the information about the collection as I said in the above comment.
R: Thank you very much for this comment. We have added the information about the population and corrected the description of population in the revised manuscript.
- Line 230, caption figure 1. Units are missing in the X and Y axis of 1A and 1B graphs. Specifically PRL should probably be PRL (cm), etc. Please use unit for Plasticity too in 1B, or please provide an explanation about Plasticity in the caption
R: Thank you very much for your careful checking on this. We have added the units for traits in the Y axis of Figure 1A and the explanation about the phenotypic plasticity in Figure 1B.
- Line 239. Apparently the authors did not plot the GWAS results for the control conditions. Why? I would welcome those graphs in the main text. On the opposite, I do not see much information in the current fig 2, I suggest to move it in supplementary.
R: Thank you very much for your suggestion. We plotted the GWAS results for two treatments and the N response value of each trait. The current Figure 2 was moved to Supplementary Figure S3. And Manhattan plots of traits which were significantly associated with corresponding SNPs under low-N condition was used as Figure 2 in the revised manuscript. In addition, we added the Manhattan plots of traits under control condition and the N response value in Supplementary Figure S4 and S5, respectively.
- The authors must comment on the fact that the root developmental stage they have targeted is very early in maize development; it is actually a time window when, in my opinion, the heterotrophic phase (eg. surviving of the seedling based on seed resources) is not even completed.
R: Thanks for this suggestion. As suggested, we discussed it in the revised manuscript Lines 518-521 as follows:
“However, it was difficult to visualize and measure root structures and their growth compared to aboveground structures. Previous studies have indicated that the hydroponic cultivation system can be used for high-throughput phenotypic screens of maize at the early stages.”
- Line 414. I would say 'often involves'
R: Thanks for your suggestion.
Round 2
Reviewer 1 Report
The authors have addressed all the concerns raised in the previous version of the manuscript and improved significantly. It can be accepted in its current form after addressing these minor comments. 1- A brief description of the methodology regarding the treatments should be provided in the abstract. 2-Please, check the provided concentrations of Ca(NO3)2 and N in CK and LN (Line 123-127 in the revised version). 3-I am not satisfied with the description regarding the treatments. Authors used LN and HN. On which basis do they categorize these levels for Ca(NO3)2 and N ? Provide some reference information to specifically select these levels.Author Response
Dear reviewer,
Thanks a lot for your consideration of our manuscript for publication in Genes pending a minor revision. In the revised version we had carefully considered the three points you had raised. Our responses to the comments point to point are presented as below, and the corresponding changes are in the text of revised manuscript.
Thanks again for your consideration of our manuscript. We would be glad to respond to any further questions and comments that you may have.
Yours sincerely,
Pengcheng Li
In response to the reviewer 1 comments: (Reviewer’s comments are marked in bold)
- A brief description of the methodology regarding the treatments should be provided in the abstract.
R: Thank you very much for your valuable comment and constructive suggestion. Maize plants were cultivated hydroponically under normal (2 mM nitrate) and low-N (0 mM nitrate) conditions. We added the description of the methodology regarding the treatments in the abstract (Line 19-20).
- Please, check the provided concentrations of Ca(NO3)2and N in CK and LN (Line 123-127 in the revised version).
R: Thank you very much for this suggestion. We have checked and revised the concentrations of Ca(NO3)2 and N in CK and LN in the revised manuscript.
- I am not satisfied with the description regarding the treatments. Authors used LN and HN. On which basis do they categorize these levels for Ca(NO3)2and N ? Provide some reference information to specifically select these levels.
R: Thank you very much for this comment. Wang et al. evaluated the effect of different levels of nitrate concentration (0.04, 0.2, 2, and 4 mM) to root morphology and N-uptake efficiency of five maize inbred lines at seedling stage (Wang et al., 2011, Journal of Plant Nutrition). The shoot dry weight and total N accumulation significantly increased above 2 mM NO3-. The 2 mM N concentration was chosen as a normal condition in our study, especially at early seedling stages. And we added the reference in the revised manuscript as suggested.